# Octyl Gallate Induces Pancreatic Ductal Adenocarcinoma Cell Apoptosis and Suppresses Endothelial-Mesenchymal Transition-Promoted M2-Macrophages, HSP90α Secretion, and Tumor Growth

**DOI:** 10.3390/cells9010091

**Published:** 2019-12-30

**Authors:** Kee Voon Chua, Chi-Shuan Fan, Chia-Chi Chen, Li-Li Chen, Shu-Chen Hsieh, Tze-Sing Huang

**Affiliations:** 1National Institute of Cancer Research, National Health Research Institutes, Miaoli 350, Taiwan; prisckv9047@gmail.com (K.V.C.); change0935693367@hotmail.com (C.-S.F.); poseking2430@nhri.org.tw (C.-C.C.); lilichen@nhri.org.tw (L.-L.C.); 2Graduate Institute of Food Science and Technology, College of Bioresources and Agriculture, National Taiwan University, Taipei 106, Taiwan; scjhsieh@ntu.edu.tw; 3Department of Biochemistry, School of Medicine, Kaohsiung Medical University, Kaohsiung 807, Taiwan; 4Ph.D. Program in Tissue Engineering and Regenerative Medicine, Biotechnology Center, National Chung Hsing University, Taichung 402, Taiwan

**Keywords:** octyl gallate, endothelial-to-mesenchymal transition, cancer-associated fibroblasts, myeloid-derived macrophages, extracellular HSP90α

## Abstract

Octyl gallate (OG) is a common antioxidant and preservative safely used in food additive and cosmetics. In this study, OG exhibited an activity to induce apoptosis in pancreatic ductal adenocarcinoma (PDAC) cells. It induced BNIP3L level and facilitated physical associations of BNIP3L with Bcl-2 as well as Bcl-X_L_ to set the mitochondrial Bax/Bak channels free for cytochrome *c* release. In addition, in vivo evaluation also showed that daily oral administration of OG was efficacious to prevent the tumor growth of PDAC cell grafts. Considering PDAC is a desmoplastic tumor consisting of many cancer-associated fibroblasts (CAFs), we further evaluated the efficacy of OG in a CAFs-involved PDAC mouse model. Endothelial-to-mesenchymal transition (EndoMT) is an important source of CAFs. The mix of EndoMT-derived CAFs with PDAC cell grafts significantly recruited myeloid-derived macrophages but prevented immune T cells. HSP90α secreted by EndoMT-derived CAFs further induced macrophage M2-polarization and more HSP90α secretion to expedite PDAC tumor growth. OG exhibited its potent efficacy against the tumor growth, M2-macrophages, and serum HSP90α level in the EndoMT-involved PDAC mouse model. CD91 and TLR4 are cell-surface receptors for extracellular HSP90α (eHSP90α). OG blocked eHSP90α–TLR4 ligation and, thus, prevented eHSP90α-induced M2-macrophages and more HSP90α secretion from macrophages and PDAC cells.

## 1. Introduction

Pancreatic ductal adenocarcinoma (PDAC) is a deadly cancer, which is usually diagnosed late and is nearly resistant to all conventional chemotherapy and radiotherapy. One of the crucial factors contributing to the aggressiveness of PDAC is the distinct and flourishing desmoplastic reaction in the tumor microenvironment [1]. Accumulating evidence has shown that stromal components such as cancer-associated fibroblasts (CAFs), myeloid-derived macrophages, and extracellular matrix can interact with cancer cells to facilitate PDAC development and malignant progression [1]. More understanding of the desmoplastic microenvironment of PDAC will be the bases for further studies on how to prevent disease progression and improve therapeutic efficacy.

With regard to CAFs, they constitute the majority of tumor stromal cells and arise from diverse resources, such as tissue-resident fibroblasts, mesenchymal stem/progenitor cells, stellate cells, and infiltrating fibrocytes [2]. The endothelial-to-mesenchymal transition (EndoMT) of endothelial cells is also an important resource contributing to 30–40% of CAFs [3]. EndoMT-derived CAFs can exhibit a potent tumor-promoting effect by secreting proteins like HSP90α, TGF-β, and soluble Jagged-1 [4,5]. Besides CAFs, myeloid-derived macrophages are abundantly present in the tumor stroma. After interacting with tumor cells and other components within the tumor microenvironment, myeloid-derived macrophages polarize to M2-type and exhibit distinct tumor-promoting activities but not tumoricidal function [6,7]. Clinically, a higher level of M2-macrophages is significantly associated with PDAC malignancy [8,9]. A tight correlation between the levels of CAF and M2-macrophage has also been revealed in colorectal cancer and PDAC tissues [5,10]. Recently, we have found that HSP90α secreted by EndoMT-derived CAFs was able to induce macrophage M2-polarization [5]. Besides M2-type markers, the extracellular HSP90α (eHSP90α) induced a feedforward loop of HSP90α secretion in macrophages, and thus a large amount of HSP90α could be expressed and secreted from eHSP90α-treated macrophages through CD91 and TLR4 receptors and the downstream MyD88-JAK2/TYK2-STAT-3 pathway [5]. Moreover, anti-HSP90α antibody exhibited a potent therapeutic efficacy against the EndoMT-promoted and M2-macrophage-involved PDAC tumor growth [5]. The discovery of agents and strategies targeting the desmoplastic microenvironment can hopefully be a new direction to develop effective PDAC therapeutics.

Octyl gallate (OG) is an octyl ester of 3,4,5-trihydroxybenzoic acid, which has the molecular formula C_15_H_22_O_5_ and molecular weight of 282.34 g/mol. It can be extracted from Chinese gall (or sumac gallnut). The gallnut is a hardened plant excretion induced by insects like gall wasps and aphids. The sumac gallnut produced from *Rhus chinensis* is a common Chinese traditional medicine used to treat coughs, diarrhea, night sweats, dysentery, and to stop intestinal and uterine bleeding. Nowadays, OG is safely used as an antioxidant and preservative in food additive and cosmetics. It is lipid-soluble and thus permeable to cell membrane [11]. Several studies have reported the chemopreventive and anti-carcinogenic effects of gallic acid and its derivatives in animal tumors or human cancer cell lines [12,13,14]. OG induced apoptosis in tumor cells and showed an anti-proliferative effect on melanoma cells [15]. A recent study also showed that OG induced mitochondrial-mediated apoptosis in the hepatocellular carcinoma cell line [16]. In this study, we evaluated the anti-PDAC efficacy of OG in vitro and in vivo and investigated the mechanism for OG-induced PDAC cell death. Furthermore, we investigated whether OG affected the interactions among cancer cells and different stromal cells and studied the underlying mechanism.

## 2. Materials and Methods

### 2.1. Cell Culture

Human PDAC cell line AsPC-1, human monocytic leukemia cell line THP-1, mouse PDAC cell line Panc 02, and mouse endothelial cell line 3B-11 were cultivated in a 37 °C and 5% CO_2_ humidified incubator with RPMI medium containing 10% fetal bovine serum (FBS) and a mixture of 100 units/mL penicillin, 100 μg/mL streptomycin, and 2 mM of l-glutamine (1 × PSG). Human PDAC cell line PANC-1 and mouse macrophage line RAW264.7 were cultivated with Dulbecco’s Modified Eagle’s Medium (DMEM) plus 10% FBS and 1 × PSG.

### 2.2. Reagents

For cell treatment, OG (Sigma-Aldrich, St. Louis, MO, USA) was dissolved in dimethyl sulfoxide (DMSO). For oral administration in mice, OG was dissolved in 40% PEG400 (Sigma-Aldrich) aqueous solution. Osteopontin (OPN; R&D Systems, Minneapolis, MN, USA) was dissolved in PBS to induce the EndoMT of 3B-11 cells (Appendix A) [4,5]. Recombinant HSP90α (rHSP90α) was purchased from Enzo Life Sciences Inc. (Farmingdale, NY, USA) and filtrated with 0.2-μm filters before use.

### 2.3. Mouse Experiments

Mouse experiments were performed in accordance with the protocols approved by the Institutional Animal Care and Use Committee of National Health Research Institutes (No.: NHRI-IACUC-106031-A). For oral administration, OG (10 mg/kg) dissolved in 40% PEG400 was administered daily to male C57BL/6 mice at 12 weeks of age. After the first 2 days of OG administration, 1 × 10^6^ Panc 02 cells were resuspended in 50 μL PBS mixed with 50 μL Matrigel and subcutaneously injected into the lower right back of each mouse. Measurement of tumor volumes was started on Day 14 post-inoculation and continued every other day with Vernier caliper. Mice were sacrificed and tumors were removed on Day 30 post-inoculation. Body weight and food intake of each mouse were recorded daily. For in vivo imaging system (IVIS), male C57BL/6 mice at 12 weeks of age were exposed to 9.5 Gy of X-ray irradiation prior to the transplantation with red fluorescent bone marrow cells (1 × 10^6^ cells per mouse) isolated from femurs of B6.Cg-Gt(ROSA)26Sortm4(ACTB-tdTomato-EGFP)Luo/Nar1 mice. After 1 week of transplantation, the mice were subcutaneously inoculated with Panc 02 (1 × 10^6^ cells), Panc 02 (1 × 10^6^ cells) plus PBS-treated 3B-11 (2.5 × 10^5^ cells, denoted as Endo), and Panc 02 (1 × 10^6^ cells) plus OPN-treated 3B-11 (2.5 × 10^5^ cells, denoted as EndoMT), respectively. The mice were subjected to IVIS analysis on Days 3, 6, and 9 post-inoculation using Xenogen IVIS^®^ Imaging System 200 with a DsRed Filter set (excitation at 710–760 nm and emission at 810–875 nm). For OG suppression of EndoMT-involved tumor growth, OG (10 mg/kg) was orally administered daily to male C57BL/6 mice at 12 weeks of age. After the first 2 days of OG administration, mice were subcutaneously inoculated with Panc 02 (1 × 10^6^ cells per mouse) or Panc-02 (1 × 10^6^ cells per mouse) plus OPN-treated 3B-11 (2.5 × 10^5^ cells per mouse, denoted as EndoMT). Mouse sera were collected weekly for HSP90α analysis. Tumor volumes were measured with Vernier caliper every three days from Day 13 after inoculation. Mice were sacrificed and tumors were removed on Day 32 post-inoculation. Tumor volumes were calculated with the formula 1/2 × length × width^2^ (in cm^3^).

### 2.4. Cell Viability Assay

To determine the inhibitory concentration of OG, cell viability assay was performed with 3-(4,5-dimethylthiazol-2-yl)-2,5-diphenyltetrazolium bromide (MTT; Sigma-Aldrich). Cells were seeded onto 96-well plates with a density of 9000 cells per well for AsPC-1, 7000 cells per well for PANC-1, and 5000 cells per well for Panc 02, and treated with OG at different concentrations for 72 h. The MTT reagent was then added to cells, and the formazan crystals were dissolved in DMSO. The optical density was measured at 540 nm.

### 2.5. Apoptosis Analysis

PANC-1, AsPC-1, and Panc 02 cells were seeded onto 10-cm dishes (2 × 10^6^ cells per dish) and treated 48 h with OG (116.1, 130.5, and 10.3 μM for PANC-1, AsPC-1, and Panc 02 cells, respectively). The treated cells were further labeled with propidium iodide (PI) and fluorescein isothiocyanate (FITC)-annexin V, according to the manufacturer’s instructions (BioLegend, San Diego, CA, USA). A minimum of 10,000 events for each sample were collected and analyzed using a FACSCalibur Flow Cytometer (BD Biosciences, San Jose, CA, USA).

### 2.6. Immunofluorescence

For cytochrome *c* and mitochondria staining, PANC-1, AsPC-1, and Panc 02 cells were seeded on ϕ12-mm round glass cover-slips (2 × 10^5^ cells per cover-slip) and treated with DMSO or OG for 24 h. Cells were pre-incubated with 200 nM of MitoTracker^®^ Orange CMTMRos for 30 min at 37 °C before harvest. Cells were fixed with 3.7% paraformaldehyde and blocked with 3% BSA in PBS for 30 min. Anti-cytochrome *c* antibody (Santa Cruz Biotechnology, Santa Cruz, CA, USA, sc-13156; 1:100) was applied for 1 h at room temperature, followed by incubation with Alexa Fluor 488 Dye (1:500) for another 1 h at room temperature. Nuclei were stained with 4′,6′-diamidino-2-phenylindole (DAPI) for 5 min at room temperature. Images were observed and analyzed using Leica TCS SP5 II confocal microscope and LAS AF Lite 4.0 software (Leica, Wetzlar, Germany).

### 2.7. Immunoblot Analysis and Antibodies

Cell lysates were prepared in cell lysis buffer consisting of 10 mM Na_2_HPO_4_, 1.8 mM KH_2_PO_4_, pH 7.4, 137 mM NaCl, 2.7 mM KCl, 1% Nonidet P-40, 0.5% deoxycholate, 0.3% SDS, 1 mM sodium orthovanadate, and 1 mM phenylmethylsulfonyl fluoride [9]. Preparation of mitochondrial and cytosolic fractions was completed as described previously [17]. Protein concentrations of samples were determined using a BCA Protein Assay Kit (Thermo Fisher Scientific, Waltham, MA, USA). Each sample (40 μg) was separated by SDS-PAGE and transferred onto a polyvinylidene difluoride membrane. The membrane was blocked in PBST (PBS containing 0.1% Tween 20) plus 5% nonfat milk for 60 min at 4 °C, followed by incubation with primary antibody overnight at 4 °C. After washing in PBST, horseradish peroxidase-conjugated secondary antibody (1:5000) was applied for 60 min. Protein band detection was performed using the chemiluminescent horseradish peroxidase detection reagent (Luminata™ Crescendo Western HRP Substrate; EMD Millipore, Billerica, MA, USA). Primary antibodies used for immunoblot analyses were listed as follows: Bcl-2 (BD Biosciences, #610539, 1:1000), Bcl-X_L_ (Santa Cruz Biotechnology, sc-8392; 1:1000), Bax (Santa Cruz Biotechnology, sc-7480; 1:1000), prohibitin (NeoMarker, Fremont, CA, USA, MS-261-P; 1:500), BNIP3 (Cell Signaling, Danvers, MA, USA, #44060S; 1:1000), BNIP-3L (Cell Signaling, #12396S; 1:1000), cytochrome *c* (Santa Cruz Biotechnology, sc-13156; 1:2000), HIF-1α (GeneTex Inc., Hsinchu City, Taiwan, GTX127309; 1:1000), Nrf2 (Santa Cruz Biotechnology, sc-722; 1:1000), p62 (Abcam, Cambridge, UK, ab-56416; 1:4000), p-STAT-3 (Epitomics, Burlingame, CA, USA, #2236; 1:500), STAT-3 (EMD Millipore, #04-1014; 1:1000), and HSP90α (GeneTex Inc., GTX109753; 1:1000).

### 2.8. Proximity Ligation Assay (PLA)

For the BNIP3L-Bcl-2/Bcl-X_L_ association study, PANC-1 and AsPC-1 cells were seeded on ϕ12-mm round glass cover-slips (2 × 10^5^ cells per cover-slip) and treated with DMSO or OG for 24 h. Cells were fixed with 3.7% paraformaldehyde and blocked with the blocking solution supplied by the Duolink in situ PLA kit (Olink Bioscience, Uppsala, Sweden) according to the manufacturer’s instructions. Cells were then incubated with anti-BNIP3L antibody (Cell Signaling, #12396S; 1:80) mixed with anti-Bcl-X_L_ antibody (Santa Cruz Biotechnology, sc-8392; 1:60) or anti-Bcl-2 antibody (BD Biosciences, #610539; 1:80) at room temperature for 1 h. After washed thrice with Tris-buffered saline plus 0.05% Tween 20, cells were successively incubated with PLA probes, ligation solution, and amplification solution in a humid chamber at 37 °C. Nuclei were counterstained with DAPI for 2 min in the dark at room temperature. Cover-slips were mounted overnight and the images were analyzed using Leica TCS SP5 II confocal microscope and LAS AF Lite 4.0 software (Leica). For the HSP90α−TLR4/CD91 association study, THP-1-derived macrophages were seeded on ϕ12-mm round glass cover-slips at a density of 2 × 10^5^ cells per cover-slip. After pre-incubating with 1% FBS-containing RPMI 1640 medium for 16 h, PBS or 15 μg/mL of rHSP90α plus DMSO or 10 μM of OG were added to macrophages for further 4 h. The subsequent PLA was performed as described above. The antibodies used were anti-HSP90α (AbD Serotec, Raleigh, NC, USA, AHP-1339; 1:80), anti-TLR4 (Santa Cruz Biotechnology, sc-8694; 1:40), and anti-CD91 (BD Biosciences, #550495; 1:80) antibodies. Quantification of signal dots was performed using Image-Pro Plus version 5.0.2 software (MediaCybernetics Inc., Silver Spring, MD, USA).

### 2.9. Flow Cytometry Analysis

Mouse tumor tissues were cut into ~1-mm^3^ pieces for further collagenase II digestion. After centrifugation and filtration, dissociated cells resuspended in PBS plus 1% FBS were incubated with primary antibodies for 60 min. Subsequently, cells were washed twice with PBS and further stained with the respective secondary antibodies for 40 min. After PBS washing, cells were immediately analyzed by Attune NxT flow cytometer (Thermo Fisher Scientific). The primary antibodies used for flow cytometry included: F4/80 (Abcam, ab-6640; 1:100), CD4 (Abcam, ab-65951; 1:100), CD8 (Abcam, ab-217344; 1:100), and INF-γ (Abcam, ab-25014; 1:100). Cells were also stained with isotype-matched control IgGs to define the background lines for gating the cell populations with F4/80^+^ROSA^+^, CD4^+^INF-γ^+^, and CD8^+^INF-γ^+^, respectively, which were further quantified by the use of Attune NxT software (Thermo Fisher Scientific).

### 2.10. Immunohistochemistry

Tumor sections with a 4-μm thickness were deparaffinized in xylene and rehydrated through a series of ethanol dilutions. Antigen retrieval was carried out by heating in 10 mM citrate buffer, pH 6.0, under high pressure for 10 min. Endogenous peroxidase activity was suppressed by adding 1% H_2_O_2_ for 15 min. The tumor sections were then blocked with 5% BSA and incubated overnight at 4 °C with antibodies against CD11b (Thermo Fisher Scientific, MA1-80091; 1:100), F4/80 (AbD Serotec, MCA497R; 1:100), CD163 (Santa Cruz Biotechnology, sc-33560; 1:80), cytokeratin 18 (CK18; Abcam, ab-668; 1:200), and α-smooth muscle actin (α-SMA; Abcam, ab-32575; 1:200). Detection was performed using the DAKO REAL EnVision Detection System (Produktionsvej 42, DK-2600 Glostrup, Denmark) and counterstained with hematoxylin.

### 2.11. ELISA

Mouse serum samples were diluted and loaded onto 96-well plates (100 μL per well). Anti-HSP90α antibody (AbD Serotec, AHP-1339; 1:1000) was added to each well and incubated at 37 °C for 1 h. After washing thrice with PBS, horseradish peroxidase-conjugated secondary antibody was added and incubated at 37 °C for another hour. The substrate 3,3′,5,5′-tetramethylbenzidine in 0.015% H_2_O_2_ was added, and the reaction proceeded at room temperature in the dark for 10 min. Finally, the reactions were stopped by 0.5 M H_2_SO_4_, and the OD was detected at 450 nm using an Infinite M200 microplate reader (TECAN, Männedorf, Switzerland). A series of concentrations of rHSP90α was prepared in 0.05 mg/mL BSA to be used as standards.

### 2.12. Preparation of Conditioned Media

To prepare the conditioned media of endothelial cells and EndoMT-derived CAFs, 3B-11 cells (2 × 10^6^ cells per 10-cm dish) were pre-incubated 16 h with 1% FBS-containing RPMI 1640 medium and then added with 0.3 μg/mL of OPN for another 24 h. After washing twice with PBS, control PBS and OPN-treated cells were incubated with 5 mL of fresh 1% FBS-containing RPMI 1640 medium for another 24 h. A dish of 1% FBS-containing RPMI 1640 medium without any cells was prepared simultaneously as control medium. The conditioned media were collected, centrifuged, filtrated with 0.45-μm filters, and designated as CTRL, Endo-CM, and EndoMT-CM, respectively. To prepare the conditioned media of the Panc 02 cells treated with control PBS, rHSP90α, CTRL, Endo-CM, or EndoMT-CM, Panc 02 cells (1 × 10^6^ cells per 6-cm dish) were pre-incubated with 1% FBS-containing RPMI 1640 medium for 16 h. The media were then added with control PBS or 15 μg/mL of rHSP90α or replaced with CTRL, Endo-CM, or EndoMT-CM in the absence or presence of 5 μM of OG. After 24 h, treated Panc 02 cells were washed twice with PBS and then incubated with 5 mL of fresh 1% FBS-containing RPMI 1640 medium for another 24 h. The media were collected, centrifuged, and filtrated with 0.45-μm filters.

### 2.13. RNA Isolation and Quantitative RT-PCR

RNA was extracted from RAW264.7 cells using TRIzol reagent (Thermo Fisher Scientific) and converted to cDNA by Tetro Reverse Transcriptase (Bioline Reagents Ltd., London, UK). The cDNA products were then used as the templates for real-time quantitative PCR (qPCR) analyses. The primers and reaction conditions were listed in Appendix A. qPCR was performed with QuantiNova SYBR Green RT-PCR Kit (Qiagen, Hilden, Germany) and StepOnePlus™ Real-Time PCR System (Thermo Fisher Scientific).

### 2.14. Statistical Analysis

Statistical analysis was performed by two-tailed unpaired student’s *t*-test using PRISM v 5.01 software (GraphPad Software Co., San Diego, CA, USA). Groups were considered significantly different when *p* < 0.05.

## 3. Results

### 3.1. OG Induces Apoptosis in PDAC Cells

To evaluate the anti-PDAC efficacy of OG, we first assayed whether OG was cytotoxic in human and mouse PDAC cell lines. According to cell viability curves obtained by MTT assays after OG treatment for 72 h, the IC_50_ values for PANC-1, AsPC-1, and Panc 02 cells were assessed as 116.1, 130.5, and 10.3 μM, respectively (Figure 1A). The cell death patterns induced by OG were further analyzed using FITC-annexin V/PI double-staining flow cytometry (Figure 1B). The cell populations undergoing apoptosis, in particular, were greater than those undergoing acute necrosis in all PANC-1, AsPC-1, and Panc 02 cells upon OG treatment. Mitochondrial cytochrome *c* was decreased in the PDAC cells treated with OG (Figure 2A). Immunoblot analyses further showed that cytochrome *c* level was decreased in mitochondrial fractions, whereas it was increased in cytosolic fractions (Figure 2B), indicating the release of mitochondrial cytochrome *c* to cytosol. Noted that BNIP3L binds to Bcl-2 or Bcl-X_L_ and facilitates Bax/Bak channels free for cytochrome *c* release, which eventually contributes to cell apoptosis. We further investigated whether OG affected the interaction of BNIP3L with Bcl-2 as well as Bcl-X_L_. PLA showed that OG drastically induced protein interactions between BNIP3L and Bcl-2 as well as Bcl-X_L_ in PDAC cells (Figure 2C). The result of immunoblot analysis also indicated that OG increased the BNIP3L protein level, which was reportedly induced by HIF-1α (Figure 2D). Indeed, a transient increase in HIF-1α protein level was observed in the PDAC cells treated with OG for 12 h. A recent study has shown that p62 binds and inhibits the VHL E3 ligase complex, which eventually leads to increased stability of HIF-1α. p62 can also activate Nrf2 by binding to KEAP1 and contributes to KEAP1 degradation. Indeed, OG increased both protein levels of p62 and Nrf2 in human PDAC cells (Figure 2E).

### 3.2. OG Inhibits the Tumor Growth of PDAC Cell Grafts in Mice

Next, we assayed whether OG exhibited anti-cancer efficacy in vivo. C57BL/6 mice were orally administered with OG (10 mg/kg) or control 40% PEG400 every day during the whole experimental period. After the first 2 days of OG administration, the mice were subcutaneously injected with Panc 02 cells, and the tumor volumes were superficially measured after 2 weeks. Tumor growth of PDAC cell grafts was inhibited by OG compared with control (Figure 3A). Mice were sacrificed on Day 30 post-inoculation and the tumors were removed. Tumor weights from the mice treated with OG were significantly smaller than those treated with control (Figure 3B). Loosened cell arrangement and decreased chromatin basophilia were also found in the tumor sections from the mice treated with OG (Figure 3C). The DAPI-stained tumor tissue sections revealed that OG treatment significantly caused chromatin condensation and nuclear fragmentation (Figure 3D). The food intake (Figure 3E) and body weight (Figure 3F) of mice were not affected by oral administration of OG.

### 3.3. Involvement of Stromal Cells in the Tumor Growth of PDAC Cell Grafts in Mice

PDAC is a desmoplastic cancer consisting of many CAFs and myeloid-derived macrophages in the tumor stroma. To investigate whether EndoMT-derived CAFs induced infiltration of myeloid cells into the tumor tissue to promote tumor growth, C57BL/6 mice were irradiated to delete bone marrow and then transplanted with the red fluorescent (i.e., ROSA^+^) bone marrow cells isolated from B6.Cg-Gt(ROSA)26Sortm4(ACTB-tdTomato-EGFP)Luo/Nar1 mice. After seven days of transplantation, the mice were subcutaneously inoculated with Panc 02, Panc 02 plus Endo, and Panc 02 plus EndoMT cell grafts, respectively. The infiltration of myeloid cells was monitored on Days 3, 6, and 9 after cell graft inoculation using the IVIS system. As shown in Figure 4A, infiltration of myeloid cells was detected in the Panc 02 plus EndoMT cell grafts as early as three days after inoculation. The myeloid cells infiltrated in the EndoMT-involved cell grafts were significantly greater, in particular on Day 6 and Day 9, compared with those in the Panc 02 alone and Panc 02 plus Endo cell grafts (Figure 4B). As expected, the tumor growth of EndoMT-involved Panc 02 cell grafts was greater than those of Panc 02 cells alone and Endo-involved Panc 02 cell grafts (Figure 4C). Mice were sacrificed on Day 40 and the tumors were removed. Consistently, tumor weights of EndoMT-involved PDAC cell grafts were significantly higher than those of Panc 02 alone or Endo-involved Panc 02 cell grafts (Figure 4D). Greater EndoMT-derived cells were detected in the tumors of Panc 02 plus EndoMT cell grafts, confirming the participation of EndoMT-derived cells in the promotion of tumor growth of Panc 02 cells (Appendix A). We further examined the tumor-infiltrating myeloid-derived cells using flow cytometry. Interestingly, myeloid-derived F4/80^+^ macrophages (i.e., F4/80^+^ROSA^+^ cells) were significantly increased (Figure 4E), whereas CD4^+^IFN-γ^+^ T cells (Figure 4F) and CD8^+^IFN-γ^+^ T cells (Figure 4G) were drastically decreased in EndoMT-involved PDAC tumors, compared with those in tumors derived from Panc 02 alone and Panc 02 plus Endo cell grafts, indicating that EndoMT-derived CAFs indeed orchestrated tumor-infiltrating myeloid-derived cells to facilitate PDAC tumor growth.

### 3.4. OG Suppresses the Tumor Growth of CAF-Involved PDAC Cell Grafts in Mice

To further investigate whether OG also inhibited CAF-involved PDAC tumor growth, C57BL/6 mice that were orally administered with OG (10 mg/kg) or control daily were subcutaneously inoculated with Panc 02 cells mixed with EndoMT-derived CAFs. The tumor volume was superficially measured after thirteen days. Tumor growth of CAF-involved PDAC cell grafts was inhibited by OG compared with control (Figure 5A). Mice were sacrificed on Day 32, and the tumors were removed. OG significantly suppressed tumor weights of CAF-involved PDAC cell grafts compared with control (Figure 5B). However, OG treatment did not obviously cause loosened cell arrangement and decreased chromatin basophilia in these CAF-involved tumor tissues (Figure 5C). There was also no significant difference in the results of immunohistochemical staining of CK18 and α-SMA (Figure 5D). Interestingly, OG treatment significantly increased the amount of F4/80^+^ macrophages, whereas it decreased the amount of CD11b^+^ myeloid cells and CD163^+^ macrophages in CAF-involved PDAC tumors (Figure 5E,F), suggesting that OG suppressed tumor-infiltrating M2-macrophages that contributed to tumor promotion. We previously reported that serum HSP90α level was highly detected in PDAC patients and PDAC-developing activated K-Ras knock-in mice. Indeed, serum HSP90α level was increased with the tumor growth of CAF-involved PDAC cell grafts in mice, and OG treatment significantly suppressed such elevation of the serum HSP90α level (Figure 5G). We have also reported that HSP90α secreted by EndoMT-derived CAFs was able to induce macrophage M2-polarization. We wondered if OG could inhibit EndoMT-CM or eHSP90α-induced macrophage M2-polarization. First, we confirmed that more HSP90α was secreted by EndoMT-derived CAFs by detecting the eHSP90α level from EndoMT-CM through immunoblot assay (Figure 6A). Next, macrophages were treated with EndoMT-CM or rHSP90α in the absence or presence of OG. As expected, both EndoMT-CM and rHSP90α repressed mRNA expressions of M1-associated TNF-α and iNOS, whereas those of M2-associated CD163, TGF-β, Arg1, and HSP90α were significantly up-regulated (Figure 6B,C). These effects were significantly antagonized by the presence of OG. Given CD91 and TLR4 are cell-surface receptors for eHSP90α, we further investigated whether OG exerted its inhibitory effect on the physical associations of eHSP90α with CD91 and TLR4. PLA showed that OG significantly suppressed eHSP90α–TLR4 but not eHSP90α–CD91 ligation (Figure 6D). Besides macrophages, the secreted HSP90α level was also increased in the Panc 02 cells treated with EndoMT-CM, and such effect was drastically inhibited by OG treatment (Figure 6E). To address whether the HSP90α secretion from Panc 02 cells was stimulated by eHSP90α, Panc 02 cells were treated with rHSP90α and then assayed the HSP90α secretion level. Indeed, rHSP90α stimulated the HSP90α secretion from Panc 02 cells, and such effect was abolished by OG treatment (Figure 6F). STAT-3 is an important regulator for the expression and secretion of HSP90α. rHSP90α induced the phosphorylation of STAT-3 in Panc 02 cells, but such a phenomenon was drastically abrogated by OG treatment (Figure 6G). These data suggested that OG prevented the binding of eHSP90α to its receptor TLR4 and thus blocked eHSP90α-induced M2-macrophages, a feedforward loop of HSP90α secretion, and PDAC tumor growth.

## 4. Discussion

The anti-cancer effect of OG on PDAC cells was suggested through a variety of assays in this study. Consistent with the result obtained from hepatocellular carcinoma cells [16], OG induced a mitochondrial-mediated apoptosis in PDAC cell lines. OG induced BNIP3L expression and facilitated the binding of BNIP3L to Bcl-2 as well as Bcl-X_L_, which can set the mitochondrial Bax/Bak channels open for cytochrome *c* release to the cytoplasm [18,19,20]. Under normoxic conditions, a transient increase in HIF-1α, which was reportedly an upstream transcription factor of BNIP3 and BNIP3L, was also observed in PDAC cells treated with OG.

OG, which has been suggested to present antioxidant properties, was involved in the Nrf2 antioxidant pathway. The protein levels of Nrf2 and its upstream regulator p62 were increased by OG. p62 is considered as an inhibitor of ubiquitin-mediated proteasomal degradation involved in several important pathways, such as the Nrf2 antioxidant pathway [21], caspase 8 activation in autophagy-related apoptotic pathway [22], and HIF-1α stabilization through VHL inhibition [23]. In agreement with this, the p62-mediated Nrf2 antioxidant pathway was induced in PDAC cells to deal with the stressful stimuli like oxidative stress.

The promising inhibitory effect of OG on PDAC tumor growth was demonstrated with a dose of 10 mg/kg/day in this study. The 2015 European Union report reevaluated that OG did not seem to raise concern regarding allergenicity, hypersensitivity, and intolerance, which further confirmed that OG can be safely consumed. According to the reports of the Scientific Committee for Food in 1976, the acceptable daily intake (ADI) of OG is 0.5 mg/kg body weight/day, based on a no observed effect level (NOEL) of 50 mg/kg body weight/day in long-term studies on rats and mice. Our study revealed that OG administration suppressed the growth of PDAC cell grafts in mice without affecting the health conditions of mice.

Besides inducing apoptosis of PDAC cells, we wondered whether OG could affect the tumor microenvironment. A mouse model was established to demonstrate whether EndoMT-derived CAFs recruited myeloid-derived cells into the PDAC tumor. Red fluorescent bone marrow cells were isolated from B6.Cg-Gt(ROSA)26Sortm4(ACTB-tdTomato-EGFP)Luo/Nar1 mice and transplanted into the wild-type C57BL/6 mice prior to inoculation of PDAC cell grafts. Our study showed that the EndoMT-promoted tumors were larger and higher amounts of myeloid-derived cells were detected within them. Higher amounts of myeloid-derived F4/80^+^ macrophages and decreased numbers of cytotoxic or effective T cells were detected in the tumors. These indicate that EndoMT-derived CAFs altered the tumor microenvironment to facilitate the PDAC tumor development.

To assess the efficacy of OG in the EndoMT-involved tumor microenvironment, EndoMT-derived CAFs were mixed with Panc 02 cells and subcutaneously injected into the mice. OG suppressed the EndoMT-involved PDAC tumor as well. An increase in F4/80^+^ macrophages and decrease in CD163^+^ macrophages were observed in the tumor sections of OG-treated mice. A decrease in the ratio of CD163^+^ macrophages to F4/80^+^ macrophages upon OG treatment suggested that OG inhibited macrophage M2-polarization or/and M2-macrophage infiltration within the tumor. Serum HSP90α level in mice was increased with the tumor growth of EndoMT cells-involved PDAC cell grafts, which was consistent with the status of PDAC-developing activated K-Ras knock-in mice and the clinical finding that eHSP90α was increased in the sera of PDAC patients [9]. OG suppressed the increase of serum HSP90α level in mice. More recently, we have reported that HSP90α secreted by EndoMT-derived CAFs was able to induce macrophage M2-polarization and more HSP90α secretion through cell-surface receptors CD91 and TLR4 and the downstream MyD88-JAK2/TYK2-STAT-3 pathway [5]. Our present study further indicated that OG blocked the eHSP90α–TLR4 ligation and, thus, inhibited EndoMT-CM and eHSP90α-induced macrophage M2-polarization and more HSP90α expression. Besides macrophages, EndoMT-derived CAFs affected PDAC cells as well. EndoMT-CM or rHSP90α also induced more HSP90α secretion from PDAC cells, and OG inhibited such phenomena. STAT-3 is an important transcriptional factor regulating the expression and secretion of HSP90α [5,9]. rHSP90α-induced STAT-3 activation in PDAC cells was drastically abolished by the presence of OG. Taken together, OG blocked eHSP90α–TLR4 ligation and thus prevented eHSP90α-induced M2-macrophages and more HSP90α secretion from macrophages and PDAC cells.

In conclusion, OG exerted an anti-PDAC effect not only by inducing mitochondrial-mediated apoptosis in PDAC cells, but also through the blocking of eHSP90α–TLR4 ligation and thus the prevention of eHSP90α-induced M2-macrophages and a feedforward loop of HSP90α secretion in macrophages and PDAC cells (Figure 7).

## Figures and Tables

**Figure 1 cells-09-00091-f001:**
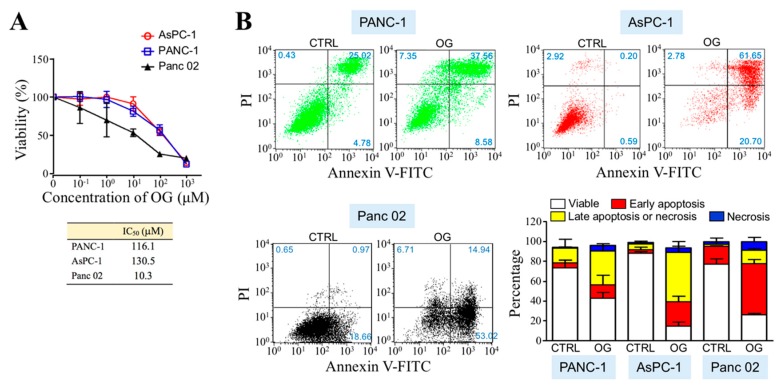
Octyl gallate (OG) induces apoptosis in pancreatic ductal adenocarcinoma (PDAC) cells. (**A**) Cytotoxic assessment of OG in PANC-1, AsPC-1, and Panc 02 cells. Cell viability curves were determined by MTT assays after OG treatment for 72 h, and IC_50_ values for each cell line were assessed using GraphPad PRISM v 5.01 software. (**B**) FITC-annexin V and PI double-staining flow cytometric analyses of the PANC-1, AsPC-1, and Panc 02 cells upon 48-h treatment of control dimethyl sulfoxide (DMSO) (CTRL) or OG (116.1, 130.5, and 10.3 μM for PANC-1, AsPC-1, and Panc 02 cells, respectively). Quantitative data are the mean ± SD of 3 independent experiments.

**Figure 2 cells-09-00091-f002:**
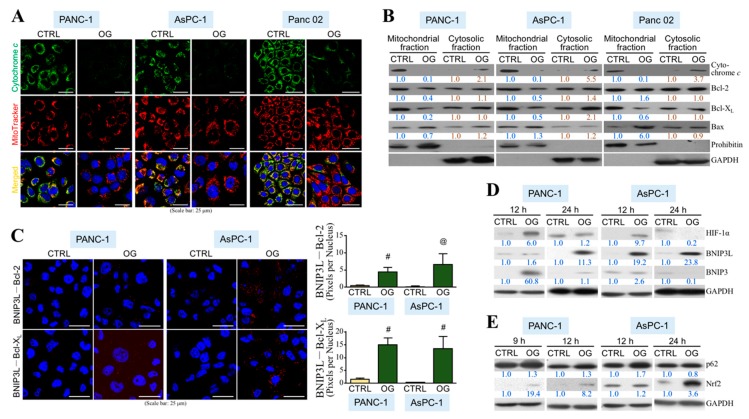
OG induces physical associations of BNIP3L with Bcl-2 and Bcl-X_L_ to facilitate mitochondrial cytochrome *c* release. (**A**) Immunofluorescent staining of cytochrome *c* and mitochondria in the PANC-1, AsPC-1, and Panc 02 cells treated 24 h with DMSO or OG (50, 50, and 5 μM for PANC-1, AsPC-1, and Panc 02 cells, respectively). Nuclei were stained with DAPI. (**B**) Immunoblot analyses of cytochrome *c*, Bcl-2, Bcl-X_L_, Bax, and prohibitin levels of the mitochondrial and cytosolic fractions isolated from the PANC-1, AsPC-1, and Panc 02 cells treated with DMSO or OG for 24 h. Band intensities were quantified using ImageJ software (National Institutes of Health, Bethesda, MD, USA). Relative levels were denoted after normalization to prohibitin (mitochondrial fraction) or GAPDH (cytosolic fraction). (**C**) Proximity ligation assay (PLA) showing physical associations of BNIP3L with Bcl-2 and Bcl-X_L_ in the PANC-1 and AsPC-1 cells treated with 50 μM of OG for 24 h. Nuclei were stained with DAPI. Quantitative data are mean ± SD of 3 independent experiments. ^@^
*p* < 0.01 and ^#^
*p* < 0.001 when compared with “CTRL” group. (**D**) Immunoblot analyses of HIF-1α, BNIP3L, and BNIP3 levels of the PANC-1 and AsPC-1 cells treated with DMSO or OG for 12 or 24 h. Relative levels were presented after normalization to GAPDH. (**E**) Immunoblot analyses of p62 and Nrf2 levels of the PANC-1 and AsPC-1 cells treated with DMSO or OG for 9, 12, or 24 h. PANC-1 cells had a higher p62 basal level compared with AsPC-1 cells and showed a little increase in response to OG treatment at any time point. Unlike PANC-1 cells, AsPC-1 cells exhibited a transient p62 induction at 12-h OG treatment.

**Figure 3 cells-09-00091-f003:**
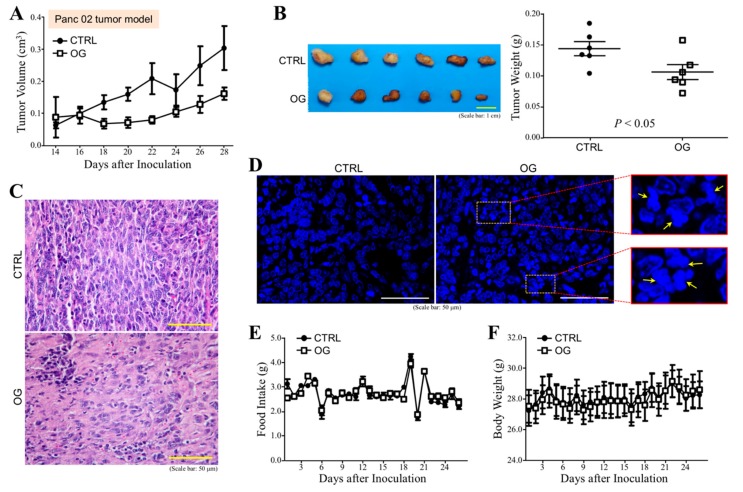
OG inhibits the tumor growth of Panc 02 cell grafts in mice. (**A**) Measurement of the superficial tumor volumes from the C57BL/6 mice subcutaneously inoculated with Panc 02 cell grafts and treated with OG or control 40% PEG400 (CTRL). The measurement was started on Day 14 post-inoculation and continued every other day using Vernier caliper with the formula 1/2 × length × width^2^. (**B**) Tumors removed from the mice described in (A) on Day 30 post-inoculation. OG treatment significantly inhibited the tumor growth of Panc 02 cell grafts. (**C**) Representative H&E-stained tumor tissue sections of control and OG-treated mice. OG treatment significantly resulted in loosened cell arrangement and decreased chromatin basophilia. (**D**) Representative DAPI-stained tumor tissue sections of control and OG-treated mice. OG treatment significantly caused chromatin condensation and nuclear fragmentation. (**E**) Food intake of the mice was monitored every day. (**F**) Body weights of the mice were also recorded daily. Data are the mean ± SEM of 6 mice.

**Figure 4 cells-09-00091-f004:**
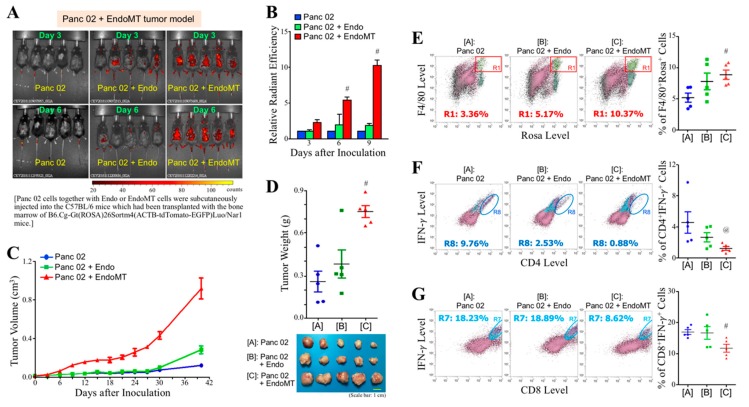
EndoMT-derived CAFs facilitate the infiltration of myeloid cells into Panc 02 cell grafts and promote tumor growth of Panc 02 cell grafts. (**A**) Representative IVIS images showing that infiltration of myeloid cells was detected in the Panc 02 + EndoMT cell grafts as early as 3 days after inoculation. C57BL/6 mice were X-ray-irradiated to delete their bone marrow and then replenished with the red fluorescent (i.e., ROSA^+^) bone marrow cells isolated from B6.Cg-Gt(ROSA)26Sortm4(ACTB-tdTomato-EGFP)Luo/Nar1 mice. These mice were further subcutaneously inoculated with 1 × 10^6^ Panc 02 cells alone or 1 × 10^6^ Panc 02 cells plus 2.5 × 10^5^ Endo or EndoMT cells. After inoculation for 3, 6, and 9 days, the mice were observed and analyzed using IVIS Imaging System. Pseudocolor scale bar: radiant efficiency (photons/sec)/(μW/cm^2^). (**B**) Quantitative results of the mice described in (A). ^#^
*p* < 0.05 when compared with “Panc 02” and “Panc 02 + Endo” groups. (**C**) Measurement of superficial tumor volume was started on Day 3 post-inoculation and continued every 3 days using Vernier caliper with the formula 1/2 × length × width^2^. (**D**) Mice were sacrificed and tumors were removed on Day 40 post-inoculation. ^#^
*p* < 0.05 when compared with “Panc 02” and “Panc 02 + Endo” groups. (**E**) Flow cytometric analysis of F4/80^+^ROSA^+^ cells (myeloid-derived macrophages) from the removed tumors. ^#^
*p* < 0.05 when compared with “Panc 02” group. (**F**) Flow cytometric analysis of CD4^+^IFN-γ^+^ T cells from the removed tumors. ^@^
*p* < 0.05 when compared with “Panc 02” group. (**G**) Flow cytometric analysis of CD8^+^IFN-γ^+^ T cells from the removed tumors. ^#^
*p* < 0.05 when compared with “Panc 02” group.

**Figure 5 cells-09-00091-f005:**
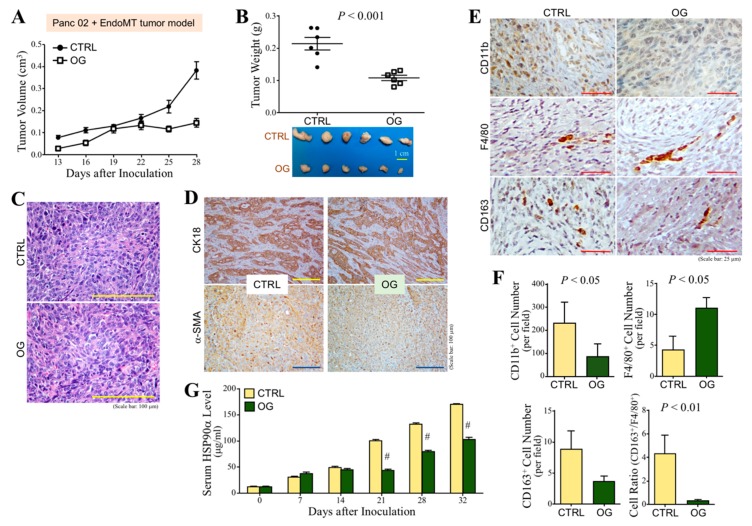
OG inhibits tumor growth, M2-type macrophages, and serum HSP90α levels in Panc 02 + EndoMT tumor model. (**A**) Measurement of the superficial tumor volumes from the C57BL/6 mice subcutaneously inoculated with Panc 02 + EndoMT cell grafts and treated with OG or control 40% PEG400 (CTRL). The measurement was started on Day 13 post-inoculation and continued every 3 days using Vernier caliper with the formula 1/2 × length × width^2^. (**B**) Tumors removed from the mice described in (A) on Day 32 post-inoculation. OG treatment significantly inhibited the tumor growth of Panc 02 + EndoMT cell grafts. (**C**) Representative H&E-stained tumor tissue sections of control and OG-treated mice. OG treatment did not obviously cause loosened cell arrangement and decreased chromatin basophilia which were observed in the Panc 02 tumor model. (**D**) Immunohistochemical staining of CK18 and α-SMA from the tumor tissues of control or OG-treated mice. OG treatment did not significantly change the CK18^+^ and α-SMA^+^ cell levels of tumor tissues. (**E**) Immunohistochemical staining of CD11b^+^, F4/80^+^, and CD163^+^ cells from the tumor tissues of control or OG-treated mice. (**F**) Quantitation of the levels of CD11b^+^, F4/80^+^, and CD163^+^ cells from the tumor tissues of control or OG-treated mice. OG treatment caused a significant increase of F4/80^+^ macrophages but decreases of CD11b^+^ myeloid cells and CD163^+^ macrophages in the tumor tissues. (**G**) Serum HSP90α levels of the mice described in (**A**). Mouse sera were collected every week and serum HSP90α levels were measured using ELISA assay. OG treatment resulted in a significant reduction of the serum HSP90α level. Data are the mean ± SD of 6 mice. ^#^
*p* < 0.001 when compared with “CTRL” group.

**Figure 6 cells-09-00091-f006:**
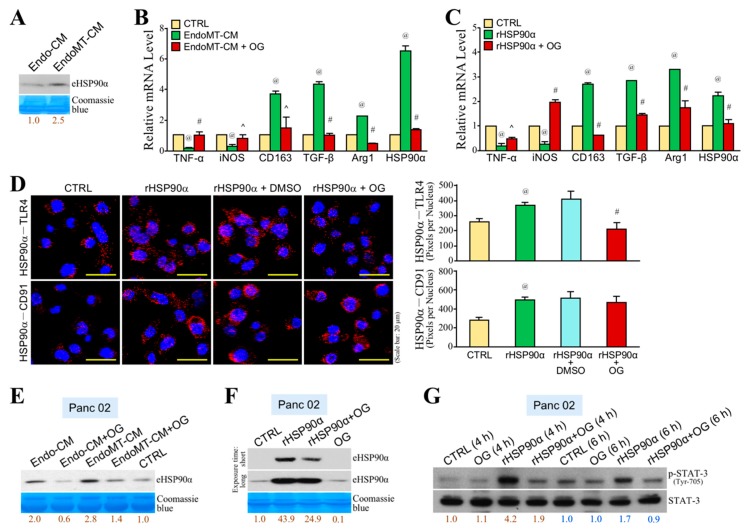
OG inhibits EndoMT-induced macrophage M2-polarization and a feedforward loop of HSP90α secretion. (**A**) Immunoblot analysis of the eHSP90α levels from Endo-CM and EndoMT-CM. EndoMT-derived cells secreted more HSP90α when compared with Endo cells. Relative eHSP90α levels were presented after quantification of protein band intensities using ImageJ software. (**B**) mRNA levels of TNF-α, iNOS, CD163, TGF-β, Arg1, and HSP90α in the RAW264.7 cells treated 24 h with control medium (CTRL), EndoMT-CM, or EndoMT-CM plus 10 μM of OG. EndoMT-CM repressed mRNA levels of M1-associated TNF-α and iNOS, whereas those of M2-associated CD163, TGF-β, Arg1, and HSP90α were significantly up-regulated. The effects were abrogated by the presence of OG. ^@^
*p* < 0.01 when compared with “CTRL” group. ^ *p* < 0.05 and ^#^
*p* < 0.01 when compared with “EndoMT-CM” group. (**C**) mRNA levels of TNF-α, iNOS, CD163, TGF-β, Arg1, and HSP90α in the RAW264.7 cells treated 24 h with PBS (CTRL) or 15 μg/mL of rHSP90α in the absence or presence of 10 μM of OG. rHSP90α treatment inhibited TNF-α and iNOS mRNA expressions but up-regulated CD163, TGF-β, Arg1, and HSP90α mRNA expression levels. The effects were antagonized by OG. ^@^
*p* < 0.01 when compared with “CTRL” group. ^ *p* < 0.05 and ^#^
*p* < 0.01 when compared with “rHSP90α” group. (**D**) PLA showing that the physical association of eHSP90α with TLR4 but not with CD91 was prevented by OG on macrophages. Nuclei were stained with DAPI. ^@^
*p* < 0.05 when compared with “CTRL” group. ^#^
*p* < 0.01 when compared with “rHSP90α + DMSO” group. (**E**) Immunoblot analysis of the eHSP90α levels from the conditioned media of the Panc 02 cells pretreated 24 h with control medium (CTRL), Endo-CM, or EndoMT-CM in the absence or presence of 5 μM of OG. The conditioned media were collected as described in Materials and Methods. The result revealed that EndoMT-CM was more than Endo-CM to enhance HSP90α secretion from Panc 02 cells, and this enhancement was inhibited by the presence of OG. (**F**) Immunoblot analysis of the eHSP90α levels from the conditioned media of the Panc 02 cells pretreated 24 h with 15 μg/mL of rHSP90α in the absence or presence of 5 μM of OG. rHSP90α treatment was able to stimulate HSP90α secretion from Panc 02 cells, which was inhibited by the presence of OG. (**G**) Immunoblot analysis of the phosphorylated and total STAT-3 levels from the Panc 02 cells treated 4 or 6 h with rHSP90α in the absence or presence of OG. Quantification of protein band intensities was performed using ImageJ software, and the relative p-STAT-3 levels were presented after normalization to total STAT-3. rHSP90α-induced STAT-3 phosphorylation (activation) was significantly suppressed by OG.

**Figure 7 cells-09-00091-f007:**
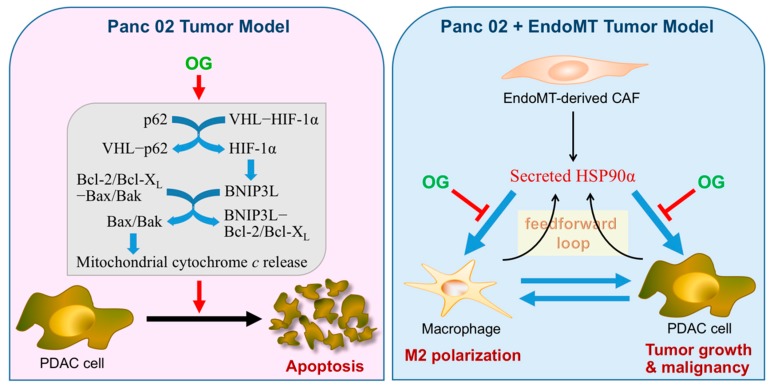
A schematic illustration summarizing our studies on the anti-PDAC mechanisms of OG. In the Panc 02 tumor model, daily oral administration of OG was efficacious to prevent the tumor growth of PDAC cell grafts. The underlying mechanism could be partly at least attributed to the ability of OG to induce a mitochondrial-mediated apoptosis in PDAC cells. OG induced p62 in PDAC cells. p62 could cause HIF-1α stabilization through binding and inhibiting the VHL E3 ligase complex, and the increased HIF-1α could further transcriptionally activate BNIP3L expression. By inducing BNIP3L expression and binding to Bcl-2 and Bcl-X_L_, OG set the mitochondrial Bax/Bak channels open for cytochrome *c* release to induce PDAC cell apoptosis. In the Panc 02 + EndoMT tumor model, OG still exhibited its potent anti-cancer efficacy even though OG did not induce tumor apoptosis as well as observed in the Panc 02 tumor model. The involvement of EndoMT-derived CAFs facilitated the recruitment of myeloid-derived macrophages into Panc 02 cell grafts. HSP90α secreted by EndoMT-derived CAFs could further induce macrophage M2-polarization and more HSP90α secretion, which resulted not only in an immunosuppressive and proangiogenic microenvironment but also creates an eHSP90α-rich condition to enhance PDAC tumor growth and malignant progression. In our study, OG blocked the binding of eHSP90α to cell-surface receptor TLR4 and thus prevented eHSP90α-induced M2-macrophages, a feedforward loop of HSP90α secretion in macrophages and PDAC cells, and PDAC tumor growth. Altogether, the anti-PDAC effects exerted by OG should be valued since OG targeted not only PDAC cells but the tumor microenvironment as well.

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
