# Peer review of "Octyl Gallate Induces Pancreatic Ductal Adenocarcinoma Cell Apoptosis and Suppresses Endothelial-Mesenchymal Transition-Promoted M2-Macrophages, HSP90α Secretion, and Tumor Growth"

_cells, 2019, doi:10.3390/cells9010091_

Round 1

Reviewer 1 Report

Dear authors:

This manuscript entitled “Octyl Gallate Suppresses Endothelial-Mesenchymal Transition-Promoted M2-Macrophage Recruitment and Tumor Growth through Inhibition of STAT-3 Mediated HSP90α Secretion” by Chua et al describes the potential anti-tumorigenic effects of the antioxidant octyl gallate on pancreatic ductal adenocarcinoma. The authors proposed a dual targeting effects  on cancer cells and their tumor microenvironment, mainly carcinoma associated fibroblasts and macrophages. The inhibition of HSP90 secretion through of STAT-3 activation presented.  While providing some new insights, it also raises several questions.

The authors include reference #5 (Fan et al), a manuscript in revision to support the secretion of proteins by EndoMT-CAF. While HSP90 and Jagged were discovered to be expressed by Endo-MT in colorectal tumors, it is not clear whether EndoMT-CAF from different tumors (in this case colorectal and pancreatic) shared the same profile. A similar statement is included to support the HSP90a secreted by CAF induces M2 polarization. It is difficult for the reviewer to evaluate these statements in the absence of a peer-reviewed publication. Perhaps Ref#5 should not be included at this time, and the authors should provide more evidence or justify.

Fig 1B (graphical) and C (quantitative) should be combined. Given that mouse Panc2 cells showed the highest sensitivity to OG, it should also be included to compare with the less sensitive human lines.

Same for Fig 1D , please include Panc2 cells and a graphical quantitation.

Fig 1E. Given that the authors are including Panc2 cells in this figure it is important to determine whether changes observed in the human lines are similar (or different) to those in mouse lines. It is not clear why the authors did not present immunoblots fro cytochrome c, Bcl-2, Bcl-XL, Bax, and prohibitin levels of the mitochondrial and cytosolic fractions for PANC-1 and AsPC-1 cells. Please include a graph quantification of the western blot.

Fig 1F also need quantification and a graphical representation (such as dot/cells). Please a include a picture at a higher magnification to show the differences.

The authors mentioned a recent study about the role of p62 that increases HIF1 stability, however there is no reference.

Fig 1G and H immunoblots need protein quantification. The authors use terms like “increased, decreased, drastically induced, leves” and others that denote quantification.

Please include in the materials and methods or the results section the rationale for using different time points in the experiments presented. In the current format it is not clear whether the reasoning for the endpoints. For example, in figure 1H for there is a 12h for PANC-1 but a 24h for AsPC-1. Did the authors noted a differential expression at these time points? Please elaborate.

Fig 2A. It is interesting that while the control group showed a steady growth rate, with OG treatment there is an initial decreased in tumor volume (perhaps due to massive induced apoptosis in Panc 02 cells?) followed by a delayed re-growth. Did the authors assessed by FACS or IHC whether the differences in tumor size is due to differential apoptotic events between both groups? Please elaborate and/or include in the discussion what is the authors’ thoughts about this observation.

Fig 2C. The stromal/epithelial ratio seems higher under OG, based on the low-magnification H&E image. The authors proposed that CAF and macrophages play a key role during PDAC progression, did the authors assessed the stromal changes in both groups? Trichrome staining or IHC analysis of putative CAF markers will be useful to determine desmoplastic lesions. Please include a high-mag H&E and quantitation of potential changes.

Please describe the gating strategy for the FACS analysis. Did the authors looked at the number of CD11b myeloid cells population? How was the data normalized? Based on the IVIS images it is expected to see a significant difference in the number of myeloid cells and F4/80+ cells.

Fig 3. Were there any histopathological differences in the tumor or stroma between the groups? Please include an H&E

Figure 4. Did the authors validated the IHC observations (F4/80 and/or CD163) using FACS analysis? How about the number of CD11b myeloid cells? Did OG induced changes in the stroma (CAF) of these tumors? Were there any histopathological difference/changes?

Fig 4 F-I. The authors state that there was differential expression in eHSP90 between the experimental groups, therefore proper immunoblot quantitation (and normalization) is required to evaluate the significance of their findings. The contribution of Panc2-derived HSP90 should be quantified (subtract from EndoMT).

Did the authors tested whether the CM of EndoMT induced p-Stat3 phosphorylation (+/- OG)?

The authors concluded that OG suppressed the infiltration of M2 macrophages by inhibiting STAT-3 activation and the subsequent inhibition of HSP90 secretion. The experiments presented partially support this conclusion. It is still not clear whether the recruitment of these myeloid cells was blocked by STAT-3 in the animal experiments. So far it seems that OG is able to suppress HSP90 secretion by tumor cells and reduce pSTAT-3. A more elegant experiment to support the authors claims should be focus on the EndoMT cells rather than the pancreatic cancer lines. The tile should be properly adjusted to support the conlcusions.

Author Response

Comment 1: “The authors include reference #5 (Fan et al), a manuscript in revision to support the secretion of proteins by EndoMT-CAF. While HSP90 and Jagged were discovered to be expressed by Endo-MT in colorectal tumors, it is not clear whether EndoMT-CAF from different tumors (in this case colorectal and pancreatic) shared the same profile. A similar statement is included to support the HSP90a secreted by CAF induces M2 polarization. It is difficult for the reviewer to evaluate these statements in the absence of a peer-reviewed publication. Perhaps Ref#5 should not be included at this time, and the authors should provide more evidence or justify.”

    Response:
Our paper reference #5 has been accepted and going to be published.  If necessary, we can provide the galley proof to assist the review of this manuscript.

Comment 2: “Fig 1B (graphical) and C (quantitative) should be combined. Given that mouse Panc2 cells showed the highest sensitivity to OG, it should also be included to compare with the less sensitive human lines.”

    Response:
The original Fig. 1B and 1C have been combined into new Figure 1B.  The data of Panc 02 cells have also been included.

Comments 3 and 4: “Same for Fig 1D, please include Panc2 cells and a graphical quantitation.”  “Fig 1E. Given that the authors are including Panc2 cells in this figure it is important to determine whether changes observed in the human lines are similar (or different) to those in mouse lines. It is not clear why the authors did not present immunoblots fro cytochrome c, Bcl-2, Bcl-XL, Bax, and prohibitin levels of the mitochondrial and cytosolic fractions for PANC-1 and AsPC-1 cells. Please include a graph quantification of the western blot.”

    Response:
New Fig. 2A and 2B have been included into the revised manuscript.  The quantitative data of western blot images have also been shown below each blot image.

Comment 5: “Fig 1F also need quantification and a graphical representation (such as dot/cells). Please a include a picture at a higher magnification to show the differences.”

    Response:
New Fig. 2C includes the quantitative graph.

Comment 6: “The authors mentioned a recent study about the role of p62 that increases HIF1 stability, however there is no reference.”

    Response:
Reference #23

Comment 7: “Fig 1G and H immunoblots need protein quantification. The authors use terms like “increased, decreased, drastically induced, leves” and others that denote quantification.”

    Response:
In new Fig. 2D and 2E, the quantitative data of western blot images have also been shown below each image.

Comment 8: “Please include in the materials and methods or the results section the rationale for using different time points in the experiments presented. In the current format it is not clear whether the reasoning for the endpoints. For example, in figure 1H for there is a 12h for PANC-1 but a 24h for AsPC-1. Did the authors noted a differential expression at these time points? Please elaborate.”

    Response:
The description“PANC-1 cells had a higher p62 basal level compared with AsPC-1 cells and showed a little increase in response to OG treatment at any time point.  Unlike PANC-1 cells, AsPC-1 cells exhibited a transient p62 induction at 12-h OG treatment.” has been added into the Fig. 2E legend (page 6 of the revised manuscript).

Comment 9: “Fig 2A. It is interesting that while the control group showed a steady growth rate, with OG treatment there is an initial decreased in tumor volume (perhaps due to massive induced apoptosis in Panc 02 cells?) followed by a delayed re-growth. Did the authors assessed by FACS or IHC whether the differences in tumor size is due to differential apoptotic events between both groups? Please elaborate and/or include in the discussion what is the authors’ thoughts about this observation.”

    Response:
The new Fig. 3D revealed that OG treatment significantly caused chromatin condensation and nuclear fragmentation (apoptotic characteristic).

Comment 10: “Fig 2C. The stromal/epithelial ratio seems higher under OG, based on the low-magnification H&E image. The authors proposed that CAF and macrophages play a key role during PDAC progression, did the authors assessed the stromal changes in both groups? Trichrome staining or IHC analysis of putative CAF markers will be useful to determine desmoplastic lesions. Please include a high-mag H&E and quantitation of potential changes.”

    Response:
The original Fig. 2C was the data of Panc 02 tumor model but not Panc 02 + EndoMT tumor model.  We have included new Fig. 3C (high-magnification H&E images) and

new Fig. 3D (DAPI staining) into the revised manuscript.

Comment 11: “Please describe the gating strategy for the FACS analysis. Did the authors looked at the number of CD11b myeloid cells population? How was the data normalized? Based on the IVIS images it is expected to see a significant difference in the number of myeloid cells and F4/80+ cells.”

    Response:
The description “Cells were also stained with isotype-matched control IgGs to define the background lines for gating the cell populations with F4/80+ROSA+, CD4+INF-gamma+, and CD8+INF-gamma+, respectively, which were further quantified by the use of CellQuest software (BD Biosciences).” has been added into the revised manuscript (page 4, lines 180-183).  The experiment in this Figure involved the bone marrow transplantation from B6.Cg-Gt(ROSA)26Sortm4(ACTB-tdTomato-EGFP)Luo/Nar1 mice (commonly called ROSA mice).  ROSA+ cells are namely myeloid cells.

Comment 12: “Fig 3. Were there any histopathological differences in the tumor or stroma between the groups? Please include an H&E.”

    Response:
Supplementary Fig. S2 has been included.

Comment 13: “Figure 4. Did the authors validated the IHC observations (F4/80 and/or CD163) using FACS analysis? How about the number of CD11b myeloid cells? Did OG induced changes in the stroma (CAF) of these tumors? Were there any histopathological difference/changes?”

    Response:
New data have been added as new Fig. 5C-5F.

Comment 14: “Fig 4 F-I. The authors state that there was differential expression in eHSP90 between the experimental groups, therefore proper immunoblot quantitation (and normalization) is required to evaluate the significance of their findings. The contribution of Panc2-derived HSP90 should be quantified (subtract from EndoMT).”

    Response:
New Fig. 6A, 6E, 6F, and 6G have been included in the revised manuscript. The preparation of conditioned media has been described in Materials and Methods.  After treatment with EndoMT-CM, Panc 02 cells were washed with PBS and then incubated 24 h with fresh medium for further collection of their secreted HSP90alpha.

Comments 15 and 16: “Did the authors tested whether the CM of EndoMT induced p-Stat3 phosphorylation (+/- OG)?” “The authors concluded that OG suppressed the infiltration of M2 macrophages by inhibiting STAT-3 activation and the subsequent inhibition of HSP90 secretion. The experiments presented partially support this conclusion. It is still not clear whether the recruitment of these myeloid cells was blocked by STAT-3 in the animal experiments. So far it seems that OG is able to suppress HSP90 secretion by tumor cells and reduce pSTAT-3. A more elegant experiment to support the authors claims should be focus on the EndoMT cells rather than the pancreatic cancer lines. The tile should be properly adjusted to support the conlcusions.”

   Response:New data in new Fig. 6B, 6C, and 6D have been included to conclude that OG blocked the binding of eHSP90alpha to its receptor TLR4 and thus prevented eHSP90alpha-induced M2-macrophages and more HSP90alpha secretion from macrophages and PDAC cells.  The title has been changed as suggested, and the text in Abstract, Results, and Discussion has also been modified in red words.  Additionally, a new Figure 7 is a schematic illustration summarizing our studies.

Reviewer 2 Report

The authors analyzed an effect of octyl gallate (OG) on cytotoxicity in pancreatic ductal adenocarcinoma (PDAC). They concluded that OG exerted anti-PDAC effect not only by inducing mitochondrial-mediated apoptosis in PDAC cells, but also through the inhibition of HSP90a secretion to suppressed tumor M2-macrophages. As OG has dual roles which influence on PDAC cells as well as tumor microenvironment, this regulation of OG is a hot topic in clinical oncology. Thus this manuscript is new and helpful for studying cancer biology. In general this manuscript is logical and interesting, and discussed a hot topic in pancreatic cancer. However the following points should be addressed.

The author should discuss as follows:

Although the author mentioned that OG has dual roles including apoptosis in PDAC, the title of this manuscript described only EndoMT–promoted-M2-macrophage recruitment. The title should be consisted of both roles.

The mechanism of OG induced suppression in PDAC is really complicated, the author should present the schema of signaling pathways of apoptosis (p62, Nrf2, KEAP1, HIF1alpha, BNIP3L, apoptosis signaling) and EndoMT–promoted-M2-macrophage recruitment (STAT3 signalng, HSP90alpha etc).

Figure 2C shows loosened cell arrangement and decreased chromatin basophilia in tumor section. This figure is too small to understand.

In Figure 3, the author used Panc 02 and EndoMT cells co-implantation mouse model. I wonder how long OPN induced EndoMT cells could continue the mesenchymal phenotype such as CAFs. The authors should show the characteristics of EndoMT cells such as CAFs markers.

In discussion, the author described OG-induced antioxidant pathway including Nrf2 signaling as well as STAT3 inhibition. The author should present the schema of signaling pathways. In addition, suppression of EndoMT–promoted-M2-macrophage recruitment is quite new phenomenon. How does OG inhibit STAT3 pathway? At least, the author should discuss how OG inhibit STAT3 phosphorylation. Did OG inhibit IL6 induced STAT3 phosphorylation or OG inhibit STAT3 phosphorylation in human PDAC?

Author Response

Comment 1: “Although the author mentioned that OG has dual roles including apoptosis in PDAC, the title of this manuscript described only EndoMT–promoted-M2-macrophage recruitment. The title should be consisted of both roles.”

    Response:
The title has been changed as suggested.

Comment 2: “The mechanism of OG induced suppression in PDAC is really complicated, the author should present the schema of signaling pathways of apoptosis (p62, Nrf2, KEAP1, HIF1alpha, BNIP3L, apoptosis signaling) and EndoMT–promoted-M2-macrophage recruitment (STAT3 signalng, HSP90alpha etc).”

    Response:
The new Figure 7 is a schematic illustration used to summarize our studies.

Comment 3: “Figure 2C shows loosened cell arrangement and decreased chromatin basophilia in tumor section. This figure is too small to understand.”

    Response:
We have included new Fig. 3C (high-magnification H&E images) andnew Fig. 3D (DAPI staining) into the revised manuscript.

Comment 4: “In Figure 3, the author used Panc 02 and EndoMT cells co-implantation mouse model. I wonder how long OPN induced EndoMT cells could continue the mesenchymal phenotype such as CAFs. The authors should show the characteristics of EndoMT cells such as CAFs markers.”

    Response:
We have included new Supplementary Fig. S1 to demonstrate that EndoMT markers were induced and lasted for > 3 days in mouse endothelial cells.

Comment 5: “In discussion, the author described OG-induced antioxidant pathway including Nrf2 signaling as well as STAT3 inhibition. The author should present the schema of signaling pathways. In addition, suppression of EndoMT–promoted-M2-macrophage recruitment is quite new phenomenon. How does OG inhibit STAT3 pathway? At least, the author should discuss how OG inhibit STAT3 phosphorylation. Did OG inhibit IL6 induced STAT3 phosphorylation or OG inhibit STAT3 phosphorylation in human PDAC?”

    Response:
We have included new data as new Fig. 6B, 6C, and 6D to conclude that OG blocked the binding of eHSP90alpha to its receptor TLR4 and thus prevented eHSP90alpha-induced M2-macrophages and more HSP90alpha secretion from macrophages and PDAC cells.  The title has been changed as suggested, and the text in Abstract, Results, and Discussion has also been modified in red words.  Additionally, a new Figure 7 is a schematic illustration summarizing our studies.

Round 2

Reviewer 2 Report

The manuscript has been much improved and is in a nice condition now.

Author Response

Thanks for your final comments.  

Some minor English changes have been done.